# Halochromic Inks Applied on Cardboard for Food Spoilage Monitorization

**DOI:** 10.3390/ma15186431

**Published:** 2022-09-16

**Authors:** Liliana Leite, Inês Boticas, Miguel Navarro, Luís Nobre, João Bessa, Fernando Cunha, Pedro Neves, Raúl Fangueiro

**Affiliations:** 1Centre for Textile Science and Technology (2C2T), University of Minho, 4800-058 Guimaraes, Portugal; 2Fibrenamics-Institute of Innovation on Fiber-based Materials and Composites, University of Minho, 4800-058 Guimaraes, Portugal; 3José Neves & Cia., Lda., Parque Industrial de Ponte 1ª Fase, Lote F, nº 277, 4801-911 Guimaraes, Portugal

**Keywords:** smart materials, intelligent packaging, halochromic inks, food packaging, food quality, food spoilage

## Abstract

Control of food spoilage is a critical concern in the current world scenario, not only to ensure the quality and safety of food but also to avoid the generation of food waste. This paper evaluates a dual-sensor strategy using six different pH indicators stamped on cardboard for the detection of spoilage in three different foods: beef, salmon, and strawberries. After function validation and formulation optimizations in the laboratory, the halochromic sensors methyl orange and bromocresol purple 2% (*w*/*v*) were stamped on cardboard and, in contact with the previously mentioned foods, were able to produce an easily perceptible signal for spoilage by changing color. Additionally, when it comes to mechanical characterization the inks showed high abrasion (>100 cycles) and adhesion resistance (>91%).

## 1. Introduction

One of the most concerning facts in today’s society is the quick population growth that has taken place in recent years. It is estimated that by the year 2050, the world population will reach 9.7 billion [1]. In line with the previous indicator, the food demand has increased 50% in the last 50 years, and it is expected to increase from 70 to 110% by 2050 [2]. Associated with this problem, there are numerous sub-problems that have a negative impact on general welfare and the environment, such as food waste and spoilage. According to the UNEP Food Waste Index 2021, around 931 million tons of food waste were produced in 2019 [3]. Furthermore, in 2016, the Food and Agriculture Organization of the United Nations stated that 70% of the total food waste in the EU arises in the household, food service, and retail sectors, and the remaining 30% in the production and processing sectors [4]. Food storage and delivery to supermarkets have increased in popularity. During those processes, due to the time and conditions in which they are handled, transported, and stored, the food products begin to dehydrate, deteriorate, and lose their color, appearance, taste, and more importantly their nutritional value [5]. The packaging process and type of package used are key points in the conservation and preservation of food quality for a longer period of time, and therefore in the decrease in food spoilage [6,7]. The EFSA defines intelligent packaging as materials/articles that “monitor the condition of packaged food or the environment surrounding the food” [8]. It is a way to reduce food waste, by extending the shelf life, monitoring freshness, and exhibiting information about its quality [6]. In addition, at the consumer level, there is a constant demand for information, namely the product history, its location, or the conditions to which the product has been subjected [6]. Among intelligent packaging, there are three common technologies, namely data carriers, indicators, and sensors [9]. Indicators determine the presence or absence of a given substance, its concentration, and/or the extent of a given reaction. Additionally, they are able to show all these variations through direct changes, such as color [10]. One critical point in the assessment of food spoilage is the presence of microorganisms and the degradation of food quality. This control can be performed indirectly, i.e., by detecting bacterial growth products, or by detecting variations in the conditions of the medium resulting from the presence of pathogens, such as changes in pH, temperature variations, or water activity. In the situations previously mentioned, the presence of microorganisms and degradation of food quality, there is a common factor: the variation of pH in the food and/or its surroundings. For example, during the spoiling process of meat and fish, there can be an increase in total volatile basic nitrogen (TVBN), which leads to a more basic pH inside the package [11,12]. A research study conducted by Kuswandi and Nurfawaidi (2017) showed the application of methyl red and bromocresol purple used separately as pH sensors for the detection of red meat spoilage [11]. In the case of milk, there is a decrease in pH due to the presence of lactic acid, a by-product of bacterial growth [13]. A study conducted with apples has shown the correlation between apple ripeness, alteration of texture, and flavor and the pH measured as a consequence of aldehyde emission. In this research, the application of a sensor based on methyl red for the detection of aldehyde was demonstrated [14]. Hence, the use of an on-package colorimetric and pH-responsive sensor is a sensitive, simple, user-friendly, and cost-effective alternative to monitor food freshness.

For the present study, the focus was to develop a solution to monitor microbiological activity as well as food degradation using functionalized inks with pH indicators directly printed on cardboard, for further application on the inside of food packages. For this a dual-sensor approach was tested, using a combination of two pH indicators with sensitivity for different areas of the pH scale. The use of two different indicators combined in the same solution allows a higher accuracy of the results, avoiding false positives [11]. The pH indicators used in this work were bromothymol blue (BB), methyl red (MR), methyl red sodium (MRs), methyl orange (MO), and bromocresol purple (BP). The cardboards functionalized with these pH indicators were able to change their color, depending on whether they came into contact with acidic or alkaline solutions. Furthermore, the cardboard printed with a double sensor exhibited the capacity to detect the spoilage of three different foods, fish, meat, and fruit. Thus, this study demonstrated the potential of pH-sensitive inks on cardboard packages as a tool for signaling food deterioration and/or microorganism proliferation.

## 2. Materials and Methods

### 2.1. Materials

Methyl red (C_15_H_15_N_3_O_2_), Methyl red sodium salt (C_15_H_15_N_3_O_2_Na), Methyl orange (C_14_H_14_N_3_NaO_3_S), bromocresol purple (C_21_H_16_Br_2_O_5_S), and bromothymol blue (C_27_H_28_Br_2_O_5_S) were purchased from Sigma-Aldrich. The polymeric base for ink preparation was provided by José Neves & Cia., Lda (Ponte, Portugal), as well as the cardboard. The filter paper, used as substrate at the beginning of this work, was purchased from Normax.

### 2.2. Preparation of pH-Sensitive Aqueous Solutions and Application on Paper Substrates

The pH-sensitive solutions were formulated exclusively thinking about their potential for food monitorization. The selection of bromothymol blue (BB), methyl red (MR), methyl red sodium (MRs), methyl orange (MO), and bromocresol purple (BP) indicators were based on previous research works [11,14,15]. Three different combinations of these indicators were tested to prepare the final indicator solutions. The first, where the indicators MR:MRs were combined in a ratio of 1:1; the second, combined MO:BP with the same ratio (1:1); and the last, which was a combination of MR:BB in a ratio of 3:2. The ratios used were also based on the previous research cited above. The preparation of the solution requires distilled water and magnetic stirring (300 rpm). The concentration of the indicators in water was 0.5, 1, and 2% (*w*/*v*). The 1 and 2% (*w*/*v*) concentrations were used to study the influence of concentration on the time-response of substrates. After 1 h of mixing, the solution was passed to a Petri dish where the filter paper was immersed for another hour. Following that, the filter paper was removed and dried at room temperature.

### 2.3. Preparation of pH-Sensitive Inks and Application on Cardboard Substrates

The conditions used for ink preparation were similar to those given above for aqueous solutions. At this stage, the aqueous base has been replaced by the polymer base. The polymer used was liquid with a varnish base and had a viscosity between 200 and 300 mPa/s, required to carry out the printing technique. Furthermore, this polymeric solution was selected in order to meet José Neves’ requirements for further industrial application by flexography. The combinations of the above-referred indicators were dissolved in the polymer maintaining vigorous agitation for 1 h. The inks were then ready to print on the cardboard substrates using stamping or flexography techniques. For each sample, one layer of ink was used, and all samples were dried at room temperature. The cardboard samples obtained can be seen in Figure 1.

### 2.4. Characterization of pH-Sensitive Aqueous Solutions and Paper Substrates for Color

Color characterization was carried out in aqueous solutions and filter paper substrates. Initially, a direct contact assay was performed. To cover the required pH range from 3 to 12, two different solutions were used. To achieve an acidic pH, a solution of 1% (*w*/*v*) of citric acid in water was made and added dropwise until reaching the desired pH. On the other hand, 1% (*w*/*v*) monosodium phosphate (NH_2_PO_4_) aqueous solution was used to obtain an alkaline pH. After achieving the required pH, the color of all samples was evaluated using the RGB color model.

The influence of concentration on color and time-response of the samples was also evaluated. As mentioned in Section 2.2 filter paper substrates were functionalized with pH-sensitive solutions in concentrations 0.5, 1, and 2% (*w*/*v*). The samples were then placed on top of Petri dishes filled with 3 mL of pure ammonia for 300 s, without direct contact. After that, RGB color coordinates were measured. ImageJ software was used to determine the RGB (red, green, and blue) values from the images. The RGB model is a combination of the different amounts of red, green, and blue primary lights to produce a variety of colors. Each of the primary colors corresponds to a value between 0 and 255. When all coordinates correspond to 0 the color black is obtained, and if they correspond to 255, the color white is achieved [16]. Image acquisition was performed in environmental conditions with a digital camera Canon PowerShot SX530 HS, with 16 megapixels resolution and 4.3 mm focal length.

### 2.5. Adhesion and Abrasion Properties of Cardboard Substrates

The determination of the abrasion resistance of printed materials consisted of using a circular cardboard sample from the area where the ink was applied with a diameter of 127 mm. The Martindale equipment was used in this article to simulate the constant abrasion that the cardboard would undergo in transportation, storage, and other normal processes. In this specific case of packaging, a printing universal paper sheet with approximately 80 g/m^2^ was used as a base pattern on which 100 abrasion cycles were performed. A cycle refers to one full rotation with a pressure of 9 kPa and a rotational frequency of 44.5 ± 2.4 min^−1^. This method was adapted from the standard ASTM D5264 [17].

The adhesion method covers the procedures for evaluating coating adhesion to substrates. The approach used was an adaptation of methodology B mentioned in standard ASTM D3359 [18]. In this sense, 2.5 cm *×* 2.5 cm squares were drawn in a test area equal to 6.25 cm^2^, resulting in a grid of 25 squares. It is important to bear that the test must be carried out in an area where the coating is as uniform as possible. Afterwards, the test area was lightly pressed with a cutting element. After the test area was well delineated, a section of tape was placed diagonally over the grid and left there for 90 s. Then, the tape was removed at an angle of 180º through a single movement. The analysis grid was evaluated before and after in ImageJ software to calculate the percentage of coating removed from the substrate.

### 2.6. Evaluation of Cardboard Substrates’ Sensitivity to Gaseous Atmospheres

To assess the sensitivity of the prepared cardboard samples to the presence of gases (acidic and alkaline), a set-up was prepared in which each sample was left for 5 min to guarantee the formation of a saturated environment. The set-up constructed consists of a combination of a flask containing 2 mL of acid or base liquid and a 250 mL glass beaker covered with parafilm (Figure 2). Since the base (ammonia) and/or acid (HCl) used are volatile, the gaseous environment is easily created. After 5 min, the color response was recorded. To understand the change in colors the RGB coordinates were measured using ImageJ software, before and after exposure. Image acquisition was performed as described in Section 2.4.

### 2.7. Cardboard Detection of Food Spoilage

After exposing the sensors to environments saturated with acidic and alkaline chemical agents, a test was run to simulate real-life conditions. The indicator methyl orange + bromocresol purple (MO:BP) was used in this test at concentrations of 1% (*w*/*v*) and 2% (*w*/*v*). These functionalized cardboard samples were then placed in an isolated environment with three different foods: 75 g of fresh salmon, 75 g of beef steak, and strawberry. The assay was performed until the pH sensor changed color or the food showed visible signs of degradation (change in smell and color). A control condition, in which the functionalized cardboard is subjected to a closed environment in the absence of food and under the same temperature and humidity conditions, was also introduced. The experiment was carried out at room temperature (approximately 20 °C). The color of the samples was evaluated on the first and last days of the trial, and RGB coordinates were obtained using ImageJ software. Image acquisition was performed as described in Section 2.4.

### 2.8. Statistical Analysis

Statistical analysis was performed to compare the color coordinates in the assays using cardboard samples. The results are presented as the average of three replicates performed, and the respective standard deviation. The results were analyzed using a one-way ANOVA test and multiple comparisons were performed using Šídák’s test. Statistical analysis was achieved using GraphPad Prism 6 Software. A critical value for significance of *p* < 0.05 was used throughout the study.

## 3. Results and Discussion

### 3.1. Measurement of RGB Coordinates of pH-Sensitive Aqueous Solutions

Considering the functional property of the mentioned indicators, after the preparation of the solutions with a concentration of 0.5% (*w*/*v*), the color change induced by changing the pH was tested. A study of the RGB color coordinates obtained for each of the prepared indicator mixture solutions was carried out, which is presented below in Figure 3.

It is possible to verify that with the variation of the pH, the RGB coordinates also change. However, it is noticeable that, for all sensors, the colors presented at some pH values are very similar, with close RGB values. This means the indicators have a similar response to those pH values, and, as seen by image representation, the color difference is not visible to the naked eye. For MR:MRs (Figure 3a), the highest variation in the RGB coordinates occurs between the values of 5 and 6, within the working range of these pH indicators [19]. The color changes observed were as expected and are due to a shift in the maximum wavelength absorbed by the compounds. In acid conditions methyl red strongly absorbs at a wavelength of approximately 515 nm, and when subjected to a basic pH, due to deprotonation, this compound has a peak at approximately 431 nm.

For the curve correspondent to MR:BB solutions the same was observed. However, in this case, a higher slope was visible from pH 5 to 6, showing a change in the color spectrum, from orange to green. This result is explained by the reference work zone of the used indicators, which are integrated into the pH range of 4.4 to 6.0 for MR and pH 6.0 to 7.6 for BB [20]. In this case, the color changes are due to methyl red’s absorption profile described above, and to bromothymol blue, which absorbs at 433 nm when in contact with an acidic solution and at 615.5 nm at an alkaline pH [21].

For MO:BP solutions, the highest variation of RGB coordinates is seen between pH 4 and 7. For a pH of 7 and 8, the colors obtained are quite identical, and for the lower end of the pH scale, the same was verified. This was to be expected since the turning point of methyl orange is from 3.1 to 4.4 [22], and the pH range where bromocresol purple changes its color is between 5.2 and 6.8 [23]. MO changes its color due to a hypsochromic shift in maximum absorption wavelength from 508 nm to 466 nm, in acid and basic conditions, respectively [24]. On the other hand, bromocresol purple strongly absorbs at a wavelength of approximately 430 nm in acidic conditions and demonstrates a shift to 590 nm in basic solutions [23].

### 3.2. Measurement of RGB Coordinates of pH-Sensitive Paper

After the preparation of the solutions, they were applied on filter paper to measure the absorption efficiency of the indicators. These samples were also subjected to an analysis of RGB components. Following the application of the solutions on filter paper, the study of the correlation between pH and color obtained by the paper samples was carried out. As can be seen in Figure 4, it was immediately noticeable for all sensors a change in the RGB coordinates after application of the acid and base solutions in the filter paper. Thus, it can be stated that the color of the substrate used will have a direct influence on the obtained final color. Although aqueous solutions have exhibited lower RGB coordinates, the variation remains similar to those obtained for paper substrates. This difference is probably due to the lower concentration of the solutions, which makes them slightly more translucent. When applied to paper, the color is much more compact and coherent, and also lighter due to the introduction of the white of the substrate. This is proved by the increase in the number of RGB coordinates, corresponding to a lighter color.

As shown in Figure 4d, it was possible to obtain a color palette with a coherent correlation between pH and color. As can be seen, by varying the pH of the solution, the color presented by the substrate also changed. This result allowed assessing the response of the pH indicators to changes in their surrounding environment. For example, looking at the paper substrate with MO:BP at pH 7 it is possible to detect a brown color, but when the sample was placed in contact with an acidic environment its color tended to change to orange. Considering these results, and since food spoilage is characterized by a change in its atmospheric pH, the relation obtained can be easily extrapolated to a label that could inform consumers about the deterioration and/or microorganisms’ presence in food. To corroborate this sentence and to optimize the sensor concentration, it was necessary to make other assays where the time and response of the substrates were measured.

### 3.3. Influence of Concentration on Color and Time-Response of the Samples

To evaluate the influence of concentration in the time-response of the substrates to gaseous environment detection tests were carried out. For this, three Petri dishes were prepared in which 3 mL of pure NH_4_ were placed. Subsequently, the paper samples were placed on the plates, without direct contact with the solution, spacing about 1 cm (Figure 5).

The RGB color model was once again applied to measure the color difference between the samples over 300 s. Thus, it was possible to confirm the indicator concentration influence on the RGB value presented by each sample. As can be seen in Figure 6, with the increase in the concentration, the RGB coordinates decrease. Since the value of the RGB coordinates decreases towards 0, corresponding to black, it is understood that the higher the solution concentration the darker the sample will become.

As shown in Figure 6, it was also found that the response time of different concentration samples is similar, since the change in their color happens at equal times. However, to the naked eye, this change is not perceptible, and because of that, it gives the illusion that the response time was different. For MR:MRs (Figure 6a), if monitoring the 2% (*w*/*v*) sample, the color changes from yellow to red appears to be faster than for a lower concentration, such as 0.5 and/or 1%. This perspective is entirely because the variation in number of the color coordinate is higher, making the detection more perceptive to the naked eye. The same was verified for the remaining combinations of indicators, MR:BB and MO:BP.

It is concluded that the concentration used has a direct influence on the color hue presented by the samples, being darker as more concentrated. However, the time-response is the same, with variation in RGB coordinates at similar times. The detection by the naked eye is the more visible the higher the concentration.

### 3.4. Adhesive and Abrasive Properties

Adhesive and abrasive properties were analyzed after cardboard functionalization with the sensors MR:MRs, MR:BB, and MO:BP. The abrasion property has a great influence on cardboard substrates, especially if they are used for packaging. When the product has high abrasion resistance and does not appear to wear out after several cycles, it is considered more robust and usually lasts longer [25]. Additionally, when it comes to packaging the inks need to have high abrasion resistance and adhesive properties to maintain the information visible to the consumer or distributor throughout its life cycle and, at the same time, preserve the aesthetic appeal of the package. The results obtained for abrasion resistance can be evaluated by visual analysis of the wear surface and/or by the mass difference before and after abrasive cycles (Figure 7A). Looking at the images presented in Figure 7A, it can be seen that the wear resistance of the cardboard samples is quite similar at 0 and 100 cycles. For the three sensors, it is not possible to visualize any changes in color or uniformity of the surface of the samples. In addition, watching the graph representation of the sample’s weight before and after cycles is shown that the achieved differences were not significant. Thus, it was concluded that the samples have high wear resistance for 100 cycles.

Adhesion is another important property in packaging products. It evaluates the resistance to a small peeling force of a coating on a cardboard surface. If the coating is weakly bonded to the substrate, then it can be easily pulled off and the adhesion is considered weak [25]. The adhesion test used follows an adaptation of the ASTM D3359 standard. In the results presented in Figure 7B), it is observed that the adhesion of the pH-sensitive ink was very similar between samples, before and after pulling off the tape. The analysis of the adhesion percentage is calculated by the difference between the removed area after the test and the total area (before the test), using ImageJ software. Therefore, it is clear that the sample with the lowest adhesion was the one functionalized with the mixture of MR:MR indicators, with a value of 91.1%. Thus, all samples present a good adhesion resistance with no significant differences in surface color and uniformity, and a good level of adhesion greater than 90%.

### 3.5. Measurement of Cardboard Sensitivity to Gaseous Atmospheres

To understand the sensitivity level of the coated cardboard with pH-sensitive inks, a test using controlled gaseous atmospheres was carried out. The results are presented in Figure 8.

When it comes to the MR:MR sample (Figure 8a), it is shown that after contact with gaseous atmospheres the values of the RGB coordinates were significantly changed, when compared with the original sample, without contact. As seen by the images presented below the graphic, starting from an initial light red color and by changing the pH to alkaline and/or acid, it is achieved a yellow and dark red color, respectively. These alterations are corroborated by the values of the individual coordinates, with the acidic conditions having lower RGB coordinates. Therefore, the sensitivity of this set of indicators to a change in pH environment was proven.

The same is true for the mixture of MR:BB indicators, where a light red color is detected before exposure, with high values for the R coordinate. However, as seen in the images presented, after exposure to an acidic and alkaline atmosphere, the colors vary to claret and green, respectively. These variations are once again accompanied by changes in coordinate values with significant alteration compared to the ones obtained for the original conditions (Figure 8b).

Once again, for the MO:BP mixture, the RGB coordinates were calculated before and after exposure to acidic and basic gas atmospheres (Figure 8c). The difference between coordinates is quite significant. As noticeable in the images below the graph, before the contact of the samples with the mentioned atmospheres, it is possible to observe a green color on the substrate, which changes to red and dark-gray when in contact with acid and alkaline atmosphere, respectively. These colors are also easily detected by changes in RGB coordinates. As can be seen, the acidic sample presented as red has a high R coordinate and lower G and B coordinate values, emphasizing the red color. On the other hand, it is possible to claim that a dark-gray sample was obtained after exposure to a high pH, since all coordinates have low absolute values close to black. In the RGB color model, the black color corresponds to the value 0, and 255 to the white. So, the brighter the sample, the higher its average RGB value; the opposite is true for dark samples that approach 0.

Thus, it is concluded that all combinations of indicators, with a concentration of 1% (*w*/*v*), respond to alkaline and acidic gaseous atmospheres.

### 3.6. Cardboard Detection of Food Spoilage

To ensure the detection of food spoilage in real-life conditions, the double sensor MO:BP was exposed to a closed environment with three different foods, meat, fish, and fruit. Since these atmospheres are not as saturated as the ones created with ammonia and HCl, to perform this assay and in line with the initial assessments, MO:BP was tested at concentrations of 1% (*w*/*v*) and 2% (*w*/*v*). The results obtained and the RGB values measured are shown in Figure 9. Statistical comparison was made for MO:BP 2% (*w*/*v*) between the samples exposed to food (0 and 2 days), and also between samples exposed to food for 2 days and the control condition after 2 days.

For all foods tested, the control RGB components are as expected, i.e., the sensor does not change its color throughout the test and the RGB coordinates are similar to the 0-day sample. This proves that the closed environment and the temperature and humidity conditions by themselves are not enough to cause a significant change in color coordinates.

In contact with fresh foods, both concentrations can express a colorimetric signal in response to degradation, as evidenced by a change in the color appearance of the cardboard and the RGB coordinates. It is important to note, however, that for the concentration of 1% (*w*/*v*), although there was a change in the color of the sensor for all three foods, the time between the start of the food spoilage process and the colorimetric response is too long since it occurred when the food already showed other signs of degradation. For salmon and beef streak samples, the detection was only possible after 3 days, which had already a noticeable color change of the foods to the naked eye, and a strong odor. Strawberry degradation, on the other hand, was detected after 10 days, and the presence of fungus was previously observed. On the contrary, for all the food samples under investigation, MO:BP 2% exhibited a total response to degradation within two days and the color change started to be seen within 24 h. At that point, there were no apparent signs of spoilage, either color, odor, and presence of microorganisms. The color obtained can be seen below the graphics. Before the contact of the samples with the atmospheres created by the foods (0 days), the RGB component shows a prominence of the red and green coordinates and a lower influence on the blue coordinate. After prolonged food exposure, the red coordinate decreases while the blue coordinate increases. This variation is corroborated by statistical analysis since all coordinates are significantly different for the samples before and after exposure to foods. The difference in color variation between the two concentrations of pH sensor was already recognizable in the preliminary work presented in Section 3.3, as the concentration has a direct influence on the samples, with darker samples being more concentrated. Furthermore, for the same time frame, the variation in the number of color coordinates is greater for higher concentrations, making detection more perceptible to the naked eye.

The color changes observed indicate a shift in the atmosphere’s pH. For all foods, this could be related to the presence and increase of volatile organic components that are usually released as a result of bacteria activity, enzymatic action, and protein decomposition, and is a sign of spoilage [26,27]. Additionally, another process usually seen during food deterioration is the release of CO_2_, related to the presence and normal metabolism of microorganisms [12,28]. For example, for beef and salmon, the type of compounds existent in the package atmosphere depends highly on storage conditions. Studies show that in an ongoing process of degradation, meat produces chemicals associated with spoilage such as esters, ketones, aldehydes, sulfur compounds, amines, and volatile fatty acids [29]. A study performed by Mikš-Krajnik et al. showed that along with other components, salmon spoilage produces trimethylamine (TMA), ethanol (EtOH), 3-methyl-1-butanol (3Met-1But), acetoin, and acetic acid [30]. On the other hand, the color change indicating fruit decomposition, in this case, strawberry, may also be associated with alterations in the volatile organic components. During the storage of strawberries, there is a variation in the levels of ethanol, ethyl acetate, and acetic acid [31].

These results and color changes, verified by the RGB color model, allow us to confirm the changes seen by the naked eye and demonstrate the development of a sensor capable of detecting the degradation of distinct classes of food, in contrast with other studies mentioned before in which the authors evaluated pH indicators and reported the identification of spoilage in only one type of food such as apples [14], salmon [12] or red meat [11]. Compared with other types of colorimetric sensors previously reported for food spoilage detection [32,33,34,35], the double sensor developed throughout this study seems easier to develop and apply to the final product, without the need for a complex procedure, different solvents, or any heat-dependent process. Furthermore, it allows a quick and direct analysis of the result, without the need for any equipment and advanced knowledge or training.

## 4. Conclusions

In this study, the use of pH-sensitive solutions was tested using different mixtures of indicators. It demonstrated the possibility of using pH sensors as indicators of food spoilage, for on-packaging applications. From the preliminary results, it was concluded that the combined ratio, concentration of the indicators, and the substrate color have a direct influence on the responsive properties of the products. When in higher concentrations the hue of the solutions was increased, and subsequently so did the visible response of the samples. However, even though the color change was observed more rapidly to the naked eye, the response of the substrates was not altered by the increased concentration since this property was still detected by RGB analysis. On the other hand, depending on the indicator mixture, the pH range and the color spectrum achieved were variable.

Regarding the cardboard, it was verified that the color obtained after printing the functional inks/solutions was affected by the color of the substrate. Concerning the analysis of the sensing capability of the inked substrates, a detection test using two distinct gas atmospheres, acidic and alkaline, was performed. All samples showed sensitivity to both environments, having a great potential for monitoring food deterioration, and/or the proliferation of microorganisms. The assay performed to simulate real-life conditions, which was performed with cardboard samples with the MO:BP sensor subjected to closed environments, disclosed the necessity to use a concentration of 2% (*w*/*v*) to ensure effective detection of food degradation before it showed visible signs of spoilage. Through this optimization, it was possible to obtain a sensor capable of detecting spoilage in three different foods, meat, fish, and fruit. Along with that, the sensor developed during this study is simple to implement at an industrial scale, has a low cost for production at a large scale, and can be directly printed on the inside of the box at the beginning of its production chain.

For future research, it would be important to perform a parallel assay using a TVBN or a CO_2_ sensor to correlate the gradual color change of the MO:BP pH sensor and the degree of food spoilage. In addition, it is relevant to evaluate the passage of this food monitoring method from a laboratory scale to an industrial scale, producing a package that could be implemented in the market. A possible printing process thought since the beginning of this research would be flexography, being then necessary to validate the maintenance of the functional and mechanical characteristics of the cardboard functionalized by this technique.

## Figures and Tables

**Figure 1 materials-15-06431-f001:**
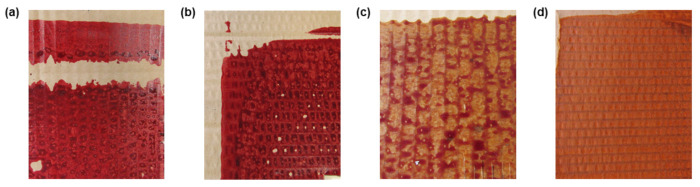
Cardboard samples after printing with pH-sensitive inks functionalized with (**a**) MR:MRs 1% (*w*/*v*), (**b**) MR:BB 1% (*w*/*v*), (**c**) MO:BP 1% (*w*/*v*), and (**d**) MO:BP 2% (*w*/*v*).

**Figure 2 materials-15-06431-f002:**
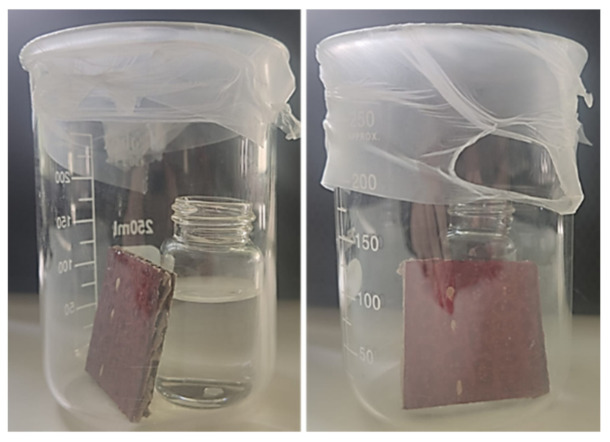
Developed set-up to evaluate the gas sensitivity of the samples.

**Figure 3 materials-15-06431-f003:**
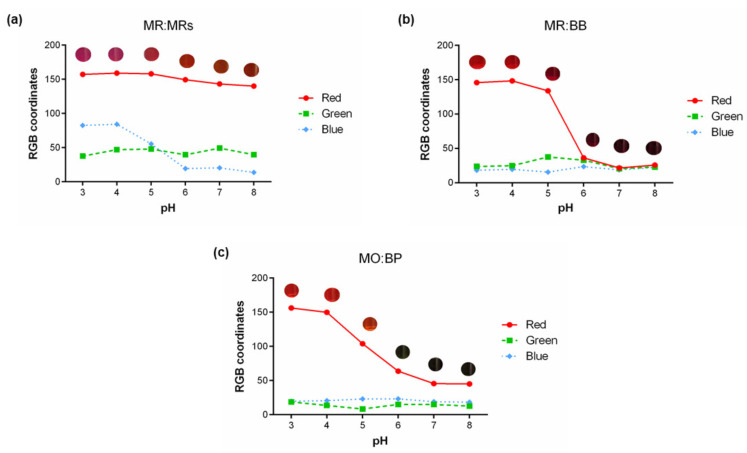
Graphical correlation of the aqueous solution color changes in response to pH. Results obtained for (**a**) MR:MRs 0.5% (*w*/*v*), (**b**) MR:BB 0.5% (*w*/*v*), and (**c**) MO:BP 0.5% (*w*/*v*) are expressed in RGB coordinates and the color obtained for every pH value is represented.

**Figure 4 materials-15-06431-f004:**
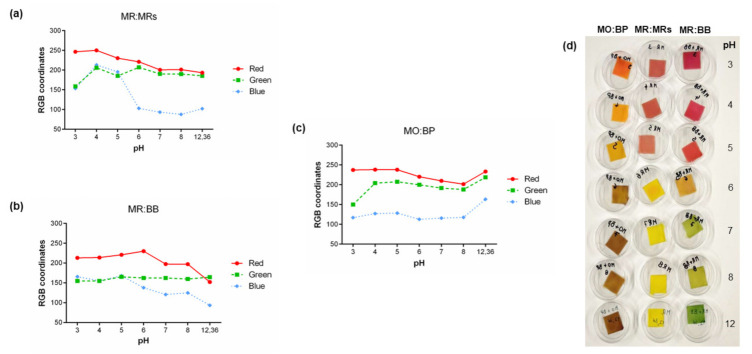
Graphical correlation of RGB color coordinates and the pH of filter paper samples functionalized with aqueous solutions of (**a**) MR:MRs, (**b**) MR:BB, and (**c**) MO:BP at a concentration of 0.5% (*w*/*v*), and (**d**) corresponding color palette.

**Figure 5 materials-15-06431-f005:**
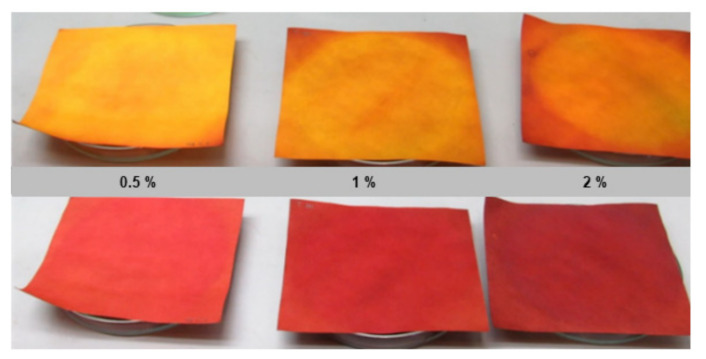
Illustration of the set-up used for detection of ammonia gases by substrates functionalized with pH indicators MR:MRs for 0.5, 1, and 2% (from left to right).

**Figure 6 materials-15-06431-f006:**
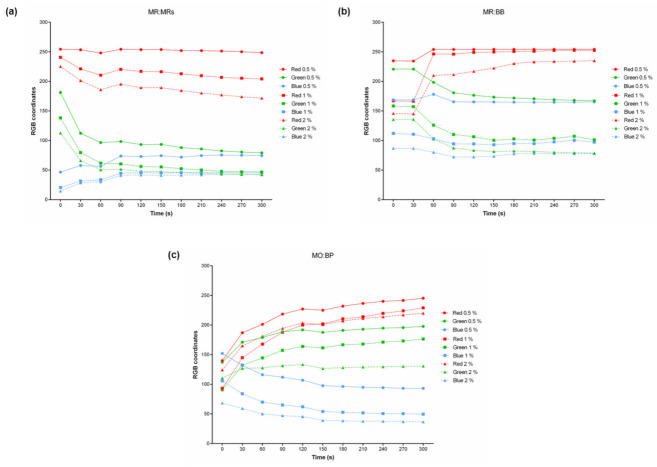
Influence of the concentration of indicators on the color change of (**a**) MR:MR-functionalized substrates, (**b**) MR:BB-functionalized substrates, and (**c**) MO:BP-functionalized substrates. Results are expressed in RGB color coordinates.

**Figure 7 materials-15-06431-f007:**
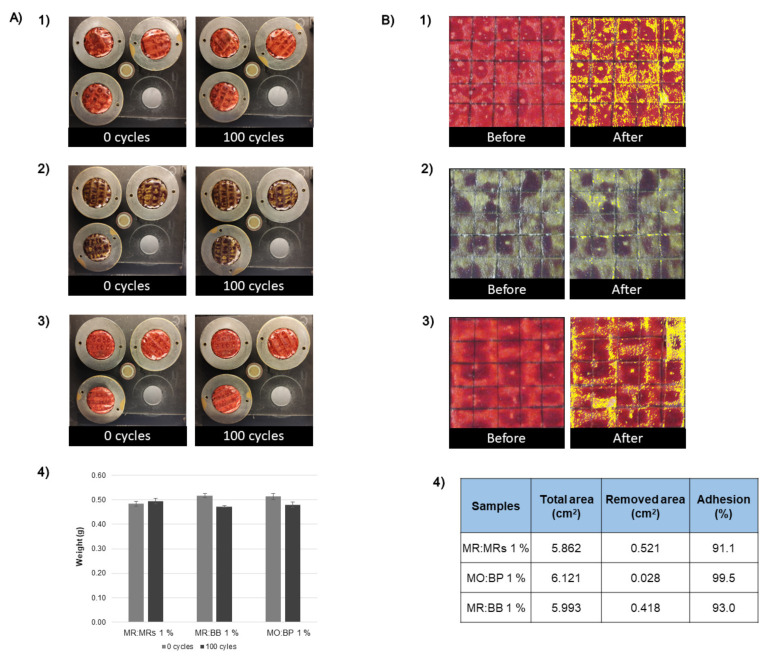
Results obtained for the (**A**) abrasion resistance of the samples (**1**) MR:MRs 1% (*w*/*v*), (**2**) MO:BP 1% (*w*/*v*), (**3**) MR:BB 1% (*w*/*v*), and (**4**) weight of samples before and after 100 abrasive cycles. Results obtained for (**B**) adhesion test of the samples (**1**) MR:MRs 1% (*w*/*v*), (**2**) MO:BP 1% (*w*/*v*), and (**3**) MR:BB 1% (*w*/*v*). The (**B4**) percentage of the adhesion was calculated by the difference between total area and removed area.

**Figure 8 materials-15-06431-f008:**
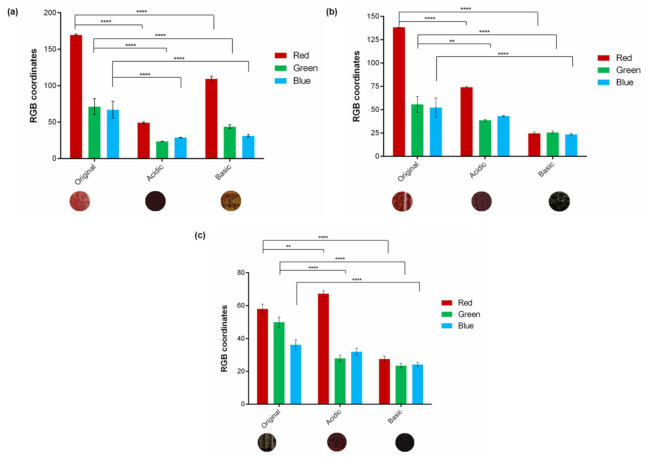
RGB coordinates and color palette presented by the samples (**a**) MR:MRs 1% (*w*/*v*), (**b**) MR:BB 1% (*w*/*v*), and (**c**) MO:BP 1% (*w*/*v*) before (original) and after exposure to acidic and basic environment (*n* = 3, ± SD), ** *p* < 0.01, **** *p* < 0.0001 (one-way ANOVA, Šídák’s test).

**Figure 9 materials-15-06431-f009:**
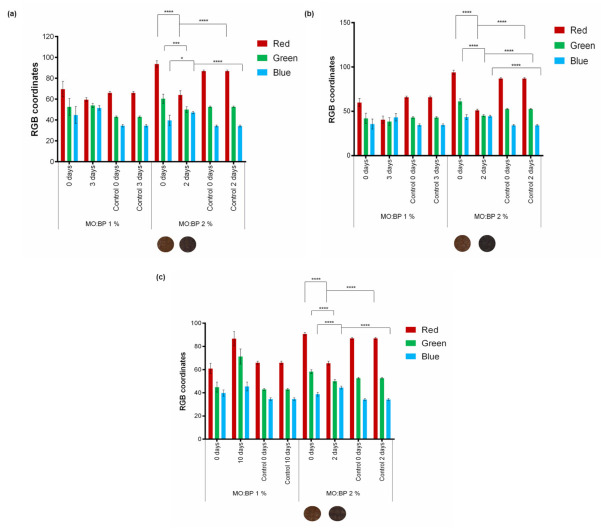
RGB coordinates presented by the MO:BP sensor at concentrations of 1% (*w*/*v*) and 2% (*w*/*v*) before and after exposure to an isolated environment within the presence of (**a**) salmon, (**b**) beef steak, and (**c**) strawberry (*n* = 3, ± SD), * *p* < 0.05, *** *p* < 0.001, **** *p* < 0.0001 (one-way ANOVA, Šídák’s test). The control refers to the sample submitted to a closed environment in the absence of food. The visual color obtained for MO:BP 2% (*w*/*v*) at 0 and 2 days is also exhibited.

## Data Availability

Not applicable.

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
