# Peer review of "Halochromic Inks Applied on Cardboard for Food Spoilage Monitorization"

_materials, 2022, doi:10.3390/ma15186431_

Round 1

Reviewer 1 Report

This is an interesting paper to read describing a study of Halochromic solutions/inks (based on mixtures of bromothymol blue (BB), methyl red (MR), methyl red sodium (MR), methyl orange (MO), and bromocresol purple (BP) in aqueous solutions transferred to paper or polymeric solutions printed on cardboard) used as pH-indicators, as an indicative of food spoilage, for packaging applications. The paper present a series of sensors subjected to different tests. The paper is interesting and I find the application on a great importance. The solution is simple and the tests are credible. Nevertheless, the analysis of results in my opinion is problematic for several reasons. The analysis are performed (qualitatively) by visual inspection of color changes (which is OK) or by an analysis using an unspecified capture instrument resulting into an image processed by ImageJ software receiving the RGB values (quantitatively). The way in which these values are used indicates that the authors are not really familiar with the RGB representation of a color. They use some quantities such as mean RGB, RGB components and a sum of red, green and blue values in percentage may be problematic. Moreover, the measurements are not supported by any other measurement instrument. Any previous information related to state of the art in this area is not presented, and the method is not compared at all with other methods. Due to this issues my recommendation to the editor is that the paper to be rejected and resubmitted with the necessary corrections. To address the problems directly: 1.    Related to the image acquisition: a.    You are not mentioned at all the details related to the image acquisition. The camera type, resolution, focal lens/distance, lightening of the sample, environmental conditions, dark room, open room, etc. The lightening and shadow can influence the RGB values. b.    In page 3 section 2. 4 you mention “When all coordinates correspond to 255 the black colour is obtained, if they correspond to 0 the white colour is achieved [15].” In my experience is vice versa. But maybe ImageJ has another representation, which is fine. But then, at the end of the page 9 you mention “… the dark-gray sample is close to black, since all coordinates have low absolute values. In RGB colour model the black colour corresponding to the value 0, and 255 to the white. So, the brighter the sample, the higher its average RGB value, the opposite is true for dark samples that approach 0.” 2.    Related to the image processing: a.    The graphs presented in Figures 2 and 3 uses the concept of “the mean RGB colour” which was not defined. What is the meaning of this? Are you using an average of Red, Green, Blue over the sample? Then should be 3 values a system of (R,G,B), which is not the case. Or you are averaging the values obtained for Red, Green, Blue? This has no sense since for example (50, 150, 80) and (150, 50, 80) have the same mean but are different colours. Please clarify this aspect. Until then Figures 2 and 3 have no meaning. b.    Figure 5 is puzzling. You have represented the RGB components and three red, green and blue curves varies is time. At the first look is Ok but the legend indicates that the red belong to the sample with 0.5 % (see the space between number and % - or in general the measurement unit), the green belong to the sample with 1 % and the blue belong to the sample with 2 %. Then what is RGB component or as in text RGB coordinates? A new quantity? Please explain. c.    Figures 7 and 8. Again you have a representation in RGB component but a normalized bar plot with red, green and blue bars can save the situation. But why are they normalized and represented in %? In this representation the colors (50, 30, 20) and (5, 3, 2) or (100, 60, 40) or (200, 120, 80) will have the same representation when clearly are different! Please represent the absolute values for RGB. 3.    As you may observe all quantitatively results are under question. 4.    What you have here is a simplified spectroscopy in Visible light of absorption or transmission/reflection light (therefore the visible spectra of the light source can be important) from your samples. Each basic color (Red, green, blue) corresponds to a well determined range of light wavelength and can be associated to a different mechanism (or chromophore) of absorption. It would be great if you can explain or associate the changes in colour in this paradigm. 5.    Other comments: a.    Page 2 you have “methyl red (MR), methyl red sodium (MR)”. Please change the acronym for the second one since is another substance. b.    Page 2 you mention “The polymeric base for ink preparation”. Please describe more the polymeric base. Is liquid? c.    Section 2.5 Please give some experimental detail for “the abrasion resistance …. test was performed using the Martindale machine.” What is a “cycle” in this context? Not all readers are familiar with all standards. d.    Figures with graphical representation. Please add the major and minor ticks. e.    Figure 2. The “bubbles” on top/bottom of the measured data are not explained in the figure legend and are not discussed. f.     Legend of the figure 3a you have 6 set of experimental data and only 3 representations. g.    Figure 5. You can change the limits of the vertical axes to represent better the variations in time. For example Fig 5a on the vertical axis you can go from 50 up to 200 or even from 80 to 180. h.    The “bubbles” on top of the bar values are not explained in the Fig. 7 and 8. i.      What is a double sensor? In which sense is double? Double sides, double components? 6.    Reference to literature: You are not mentioned (explicitly) other results similarly to yours in this direction. Is this the first paper? If not (and your last idea before conclusion is “…, in contrast with other studies that report the identification of spoilage in only one type of food.”) you should have a paragraph in introduction related to the performances of other study and in discussion other studies with yours. 7.    Related to the evaluation of your tests a.    You are using your sensors and visual inspection for the assessment of food spoilage. For a credible test you should use also some other (validated) sensors or analysis. A pH sensor or CO2 is welcomed. b.    Food quality can be another support analysis which can assess the sensitivity of your method. 8.    Related to practicability: a.    There are already various types of pH paper strips, pH sensors. Why should be used yours? b.    In the direct contact I agree that is functioning. I was particularly interested in the results presented in Fig. 4. Remotely detection or by penetrating the cardboard. The regular arrangement is that the food is inside a package and inked printed information lies outside. Then is a barrier between the food product and sensor. How you can solve this one. Please comment.

Author Response

Dear Reviewer,

First, the authors would like to thank you for your comments and suggestions. All of them were taken into consideration and the respective changes were made. Additionally, the answers to those comments are presented below.

This is an interesting paper to read describing a study of Halochromic solutions/inks (based on mixtures of bromothymol blue (BB), methyl red (MR), methyl red sodium (MR), methyl orange (MO), and bromocresol purple (BP) in aqueous solutions transferred to paper or polymeric solutions printed on cardboard) used as pH-indicators, as an indicative of food spoilage, for packaging applications. The paper present a series of sensors subjected to different tests. The paper is interesting and I find the application on a great importance. The solution is simple and the tests are credible. Nevertheless, the analysis of results in my opinion is problematic for several reasons. The analysis are performed (qualitatively) by visual inspection of color changes (which is OK) or by an analysis using an unspecified capture instrument resulting into an image processed by ImageJ software receiving the RGB values (quantitatively). The way in which these values are used indicates that the authors are not really familiar with the RGB representation of a color. They use some quantities such as mean RGB, RGB components and a sum of red, green and blue values in percentage may be problematic. Moreover, the measurements are not supported by any other measurement instrument. Any previous information related to state of the art in this area is not presented, and the method is not compared at all with other methods. Due to this issues my recommendation to the editor is that the paper to be rejected and resubmitted with the necessary corrections. To address the problems directly: 

Point 1: Related to the image acquisition: 

a. You are not mentioned at all the details related to the image acquisition. The camera type, resolution, focal lens/distance, lightening of the sample, environmental conditions, dark room, open room, etc. The lightening and shadow can influence the RGB values.

Response 1a: The authors would like to thank you for your comment. The details about image acquisition were added to section 2.4 page 5 (lines 107-108).

b. In page 3 section 2. 4 you mention “When all coordinates correspond to 255 the black colour is obtained, if they correspond to 0 the white colour is achieved [15].” In my experience is vice versa. But maybe ImageJ has another representation, which is fine. But then, at the end of the page 9 you mention “… the dark-gray sample is close to black, since all coordinates have low absolute values. In RGB colour model the black colour corresponding to the value 0, and 255 to the white. So, the brighter the sample, the higher its average RGB value, the opposite is true for dark samples that approach 0.” 

Response 1b: The authors would like to thank you for your comment. In fact, the mentioned phrase is not correct, the coordinates 0 correspond to black and 255 to white. The affirmation was altered as shown in the manuscript (lines 105-106).

Point 2: Related to the image processing: 

a. The graphs presented in Figures 2 and 3 uses the concept of “the mean RGB colour” which was not defined. What is the meaning of this? Are you using an average of Red, Green, Blue over the sample? Then should be 3 values a system of (R,G,B), which is not the case. Or you are averaging the values obtained for Red, Green, Blue? This has no sense since for example (50, 150, 80) and (150, 50, 80) have the same mean but are different colours. Please clarify this aspect. Until then Figures 2 and 3 have no meaning. 

Response 2a: The authors would like to thank you for your comment. The mean of RGB colour refers to the summation of the three coordinates, and this value is then divided by three. The results regarding pH-sensitive aqueous solutions and their application on paper substrates (paper-filter) were done as a preliminary test, with the aim of assessing if all three pH sensors selected were promising for further application on cardboard. With that in mind, when presenting the mean RGB value the authors intended to show the total colour change when comparing the initial and final colours, despite what exact colour was obtained. However, and because we understand your point and question, the figures were altered and the single values of R, G and B are now demonstrated on pages 7 and 10 as Figure 3 and Figure 4.

b. Figure 5 is puzzling. You have represented the RGB components and three red, green and blue curves varies is time. At the first look is Ok but the legend indicates that the red belong to the sample with 0.5 % (see the space between number and % - or in general the measurement unit), the green belong to the sample with 1 % and the blue belong to the sample with 2 %. Then what is RGB component or as in text RGB coordinates? A new quantity? Please explain. 

Response 2b: The authors would like to thank you for your comment. Figure 5 was again a representation of the mean RGB value in correlation with the sensor concentration, in order to study the influence of concentration on colour and time-response of the samples. Like in the data regarding the previous comment, the aim was to clarify if the concentration had a substantial impact on the time-response of our sensors and, consequently, on the final colour exhibited by the filter paper.  However, in relation to this observation and as done for the previous comment, the authors have altered the images in order to show the single values of R, G and B, as well as the comparison between the three concentrations tested (0.5 %, 1 % and 2 %) for all threes pH-sensors. The alterations can be seen on page 12 Figure 6.

c. Figures 7 and 8. Again you have a representation in RGB component but a normalized bar plot with red, green and blue bars can save the situation. But why are they normalized and represented in %? In this representation the colors (50, 30, 20) and (5, 3, 2) or (100, 60, 40) or (200, 120, 80) will have the same representation when clearly are different! Please represent the absolute values for RGB. 

Response 2c: The authors would like to thank you for your comment. The graphics have been rearranged and absolute RGB values are now represented. The alterations can be seen on page 18 Figure 8 and page 20 Figure 9.  Apart from representing separate values of RGB, for these sets of data, the authors have performed statistical analysis for better discussion of the results and clarification. In figure 8 this analysis was for the purpose of attesting the differences in the red, green and blue coordinates when comparing the original condition, and the conditions where the samples are exposed to an acidic and alkaline environment. In figure 9 the authors intend to show the statistical difference in the red, green and blue coordinates for the sample with the highest sensor concentration (2 %) since it was the one that demonstrated a better result.

Point 3: As you may observe all quantitatively results are under question. 

Point 4:    What you have here is a simplified spectroscopy in Visible light of absorption or transmission/reflection light (therefore the visible spectra of the light source can be important) from your samples. Each basic color (Red, green, blue) corresponds to a well determined range of light wavelength and can be associated to a different mechanism (or chromophore) of absorption. It would be great if you can explain or associate the changes in colour in this paradigm. 

Response 4: The authors would like to thank you for your comment. This association was introduced and discussed in section 3.1 (lines 179-199).

Point 5:    Other comments: 

a. Page 2 you have “methyl red (MR), methyl red sodium (MR)”. Please change the acronym for the second one since is another substance. 

Response 5a: The authors would like to thank you for your observation. As shown in the manuscript, the acronym for methyl red sodium was changed to MRs.

b. Page 2 you mention “The polymeric base for ink preparation”. Please describe more the polymeric base. Is liquid?

Response 5b: The authors would like to thank you for your question. The polymeric base was liquid with a varnish base, it was selected in order to meet José Neves requirements for further industrial application by flexography. This information was added to section 2.3 page 4 (lines 78-80).

c. Section 2.5 Please give some experimental detail for “the abrasion resistance …. test was performed using the Martindale machine.” What is a “cycle” in this context? Not all readers are familiar with all standards.

Response 5c: The authors would like to thank you for your suggestion. The Martindale equipment was used in this article to simulate the constant abrasion that the cardboard would undergo in transportation, storage, and others. A cycle refers to one full rotation against a universal paper sheet. In this case, the cardboard was subjected to 100 circular continuous movements with a pressure of 9 kPa and a rotational frequency of 44,5 ± 2,4 min–1. This information and description were added to section 2.5 page 5 (lines 113-117).

d. Figures with graphical representation. Please add the major and minor ticks. 

Response 5d: The authors would like to thank you for your suggestion. Major and minor ticks were added to Figures 8 and 9.

e. Figure 2. The “bubbles” on top/bottom of the measured data are not explained in the figure legend and are not discussed.

Response 5e:  The authors would like to thank you for your observation. The bubbles are photos of the samples and are there to allow visual correlation to the RGB analysis. This is now explained in the figure legend and discussed throughout the manuscript.

f. Legend of the figure 3a you have 6 set of experimental data and only 3 representations.

Response 5f: The authors would like to thank you for your comment. Since the figure was changed according to previous comments, the legend of the figure has been updated correctly in correlation to the new data exposed. This data is now represented in the manuscript on page 10 as Figure 4.

g. Figure 5. You can change the limits of the vertical axes to represent better the variations in time. For example Fig 5a on the vertical axis you can go from 50 up to 200 or even from 80 to 180. 

Response 5g: The authors would like to thank you for your comment. The mentioned figure was changed in order to represent the absolute values of RGB coordinates, and therefore the limits were changed accordingly. All vertical axes have the same limits in order to allow a better comparison between the three pH sensors tested. This data is now represented in the manuscript on page 12 as Figure 5.

h. The “bubbles” on top of the bar values are not explained in the Fig. 7 and 8.

Response 5h: The authors would like to thank you for your observation. The bubbles are photos of the samples and are there to allow correlation to the RGB analysis. This is now explained in the figures’ legends and discussed in the manuscript. The alterations can be seen throughout section 3.5 and on page 18 Figure 8, and throughout section 3.6 and on page 20 Figure 9.

i. What is a double sensor? In which sense is double? Double sides, double components?

Response 5i: The authors would like to thank you for your comment and questions. The article uses the expression “double sensor” to define two different pH indicators (and therefore compounds) combined in the same solution to form one sensor. The meaning of this expression is now clarified on page 3 (lines 42-44).

Point 6:    Reference to literature: You are not mentioned (explicitly) other results similarly to yours in this direction. Is this the first paper? If not (and your last idea before conclusion is “…, in contrast with other studies that report the identification of spoilage in only one type of food.”) you should have a paragraph in introduction related to the performances of other study and in discussion other studies with yours. 

Response 6: The authors would like to thank you for your comment. Alterations were made in the introduction in section 1 page 2 (lines 31-37) and in the discussion in section 3.6 pages 21 and 22 (lines 404-410).

Point 7:    Related to the evaluation of your tests 

a. You are using your sensors and visual inspection for the assessment of food spoilage. For a credible test you should use also some other (validated) sensors or analysis. A pH sensor or CO2 is welcomed. 

b. Food quality can be another support analysis which can assess the sensitivity of your method. 

Response 7: The authors thank you for your suggestion. Although the authors agree that the tests mentioned are relevant, those were not within the scope of this study since the main objective was the optimization of a sensor capable of being printed on cardboard and the maintenance of its functionality. Furthermore, the spoilage of beef, salmon and strawberries and the type of compounds released are already documented in other studies cited in the manuscript (lines 388-401). However, due to the importance of such analysis, those tests were mentioned in section 4 page 22 (lines 435-436) for future investigation.

Point 8:    Related to practicability:

a. There are already various types of pH paper strips, pH sensors. Why should be used yours? 

Response 8a. The authors would like to thank you for your comment. The advantages of our pH sensor were added to section 4 page 22 (line 432-434).

b. In the direct contact I agree that is functioning. I was particularly interested in the results presented in Fig. 4. Remotely detection or by penetrating the cardboard. The regular arrangement is that the food is inside a package and inked printed information lies outside. Then is a barrier between the food product and sensor. How you can solve this one. Please comment.

Response 8b. The authors would like to thank you for your comment. Colour characterization was performed throughout the study in aqueous solutions, filter paper substrates and cardboard. To test pH-sensitive aqueous solutions, we performed direct contact with solutions with different ranges of pH. After that, the same was done with filter paper substrates. However, when testing the influence of concentration on colour and time-response of the filter paper samples (results presented in Figure 6), the filter paper was placed on top of plates with pure ammonia, without direct contact.  For cardboard, the same was done, i.e the samples were subjected to gaseous environments without direct contact. To better clarify those points, since it may open room for confusion, sections 2.4 and 2.6 were altered. In section 2.4 the authors clarified the methods used for aqueous solutions and filter paper substrates and in section 2.6 the authors described the methods for colour characterization of cardboard samples.

Furthermore, in the case of this pH sensor applied on cardboard, the ink will be printed on the inside of the package so it will be in the same atmosphere as the food product and we will not have a barrier between the food and the sensor. This information was clarified in section 1 page 3 (lines 40-41). In future commercial packages, to allow the consumer to see the indicator, the box could have a small transparent zone. 

Please find attached the cover letter with the details of the revisions.

Reviewer 2 Report

Dear MDPI Materials Editor. I have carefully reviewed the manuscript entitled "Halochromic inks applied on cardboard for food spoilage monitoring". The manuscript is important in your area so I consider it suitable for publication in your journal after the authors make the following comments. My decision is minor review

Comments:

1. Include in the introduction for its improvement something similar to the following information and cite the following article

The storage of food and its arrival in supermarkets has received a greater demand. During the time and conditions that food goes through in handling, transportation and storage, the products begin to dehydrate, deteriorate, lose their color and appearance, taste, and more importantly their nutritional value.  (Physicochemical, structural, mechanical and antioxidant properties of zein films incorporated with no-ultrafiltred and ultrafiltered betalains extract from the beetroot (Beta vulgaris) bagasse with potential application as active food packaging. Journal of Food Engineering, 111153).

2. Include images of the sensors in the evaluated products if they are available for greater visualization.

3. Give future perspectives within conclusions to see what else follows in relation to your research work

4. Further discuss the results with previous research in the area of food conservation

Author Response

Dear Reviewer,

First, the authors would like to thank you for your comments and suggestions. All of them were taken into consideration and the respective changes were made. Additionally, the answers to those comments are presented below.

Dear MDPI Materials Editor. I have carefully reviewed the manuscript entitled "Halochromic inks applied on cardboard for food spoilage monitoring". The manuscript is important in your area so I consider it suitable for publication in your journal after the authors make the following comments. My decision is minor review

Point 1: Include in the introduction for its improvement something similar to the following information and cite the following article

The storage of food and its arrival in supermarkets has received a greater demand. During the time and conditions that food goes through in handling, transportation and storage, the products begin to dehydrate, deteriorate, lose their color and appearance, taste, and more importantly their nutritional value.  (Physicochemical, structural, mechanical and antioxidant properties of zein films incorporated with no-ultrafiltred and ultrafiltered betalains extract from the beetroot (Beta vulgaris) bagasse with potential application as active food packaging. Journal of Food Engineering, 111153).

Response 1: The authors thank you for your suggestion. The information was inserted in the introduction in lines 10-13, along with the citation suggested.

Point 2:  Include images of the sensors in the evaluated products if they are available for greater visualization.

Response 2: The authors thank you for your comment. At this stage, at a small scale and to assess their functionality, the sensors were stamped on cardboard pieces regularly used for packages. The photos of the products/cardboard samples obtained were included in the manuscript in section 2.3 page 4 as Figure 1. 

Point 3:  Give future perspectives within conclusions to see what else follows in relation to your research work.

Response 3: The authors thank you for your suggestion. Future perspectives were added to section 4 page 16 (lines 435-440).

Point 4:  Further discuss the results with previous research in the area of food conservation.

Response 4: The authors would like to thank you for your comment. Further discussion was added particularly in the results presented in section 3.6 pages 21 and 22 (lines 404-410).

Please find attached the cover letter with the details of the revisions.

Round 2

Reviewer 1 Report

Thank you for considering my suggestions!